# The Effects of Genotype × Environment on Physicochemical and Sensory Properties and Differences of Volatile Organic Compounds of Three Rice Types (*Oryza sativa* L.)

**DOI:** 10.3390/foods12163108

**Published:** 2023-08-18

**Authors:** Jing Yu, Dawei Zhu, Xin Zheng, Liangliang Shao, Changyun Fang, Qing Yan, Linping Zhang, Yebo Qin, Yafang Shao

**Affiliations:** 1China National Rice Research Institute, Hangzhou 310006, China; yjingyx@163.com (J.Y.); nxyzdw@163.com (D.Z.); cindy-132@163.com (X.Z.); fcyst88@163.com (C.F.); yanqing@caas.cn (Q.Y.); zhanglinping@caas.cn (L.Z.); 2Grain and Oil Product Quality Inspection Center of Zhejiang Province, Hangzhou 310012, China; shaolly@163.com; 3Argo–Technical Extension Service Center of Zhejiang Province, Hangzhou 310005, China; qinyebo@126.com

**Keywords:** pasting viscosity, sensory evaluation, flavor fingerprint, planting environment, genotype, Mantel test, principal component analysis

## Abstract

Understanding the effects of genotype, environment and their interactions on rice quality is of great importance for rice breeding and cultivation. In this study, six rice varieties with two *indica*, two *japonica* and two *indica–japonica* types of rice were selected and planted at ten locations in Zhejiang Province to investigate the genotype (G) × environment (E) on physicochemical and sensory properties and the differences of volatile organic compounds (VOCs) among the three types of rice. Analysis of variances showed that apparent amylose content (AC), total protein content (PC), alkali spreading value (ASV), RVA profiles, and appearance (ACR), palatability (PCR), and sensory evaluation value (SEV) of cooked rice and texture of cooled cooked rice (TCCR) were mainly affected by genotypic variation, whereas the smell of cooked rice (SCR) was mainly affected by environment (*p* < 0.05). The G × E effect was significant for most parameters. The weather in the middle and late periods of filling had important effects on the formation of rice quality, especially on setback (SB) and pasting temperature (PT) (*p* < 0.01). They were negatively correlated with the texture of cooked rice (TCR) and SEV (*p* < 0.05). Peak viscosity (PV) and breakdown (BD) were positively related to the sensory evaluation parameters (*p* < 0.01) and could be used to predict cooked rice quality. A total of 59 VOCs were detected, and *indica, japonica* and *indica–japonica* had 9, 6 and 19 characteristic compounds, respectively. The principal component analysis showed that the physicochemical and sensory properties and VOCs of *indica–japonica* rice were more stable than those of *indica* and *japonica* rice at ten locations in Zhejiang Province. It is helpful for rice breeders to understand how the environment affects the physicochemical, sensory properties and VOCs of the three rice types, and it is also important for food enterprises to provide rice products with stable quality.

## 1. Introduction

Rice (*Oryza sativa* L.) is one of the most important staple foods, feeding almost half of the world’s population. In recent decades, thanks to technological innovations such as dwarf breeding, heterosis utilization and super rice breeding, rice production has increased significantly, basically meeting the growing demand for rice consumption in China [1]. As a naturally gluten-free grain, rice is one of the best sources of carbohydrates for gluten-intolerant consumers [2]. In recent years, the quality of rice has received a lot of attention due to the increasing socio-economic status of consumers [3] and their awareness of the sustainability of rice production relating to food safety and health [4]. Therefore, the production of high-quality rice is very important in ensuring consumer acceptance of rice in the market [5].

Starch and protein are the main chemical components which make up about 85% and 10% of the dry weight of polished rice, respectively. For consumers, rice quality may be perceived by intrinsic characteristics based on aroma, appearance, taste and texture [3]. Rice aroma is associated with the compositions of volatile organic compounds (VOCs), including aldehydes, alcohols, ketones, acids, hydrocarbons, esters, heterocyclic compounds, and so on [6,7]. It is reported that starch properties and protein content influence the taste and texture of cooked rice [8], mainly because protein can interact with starch molecules through hydrogen bonds to affect the gelatinization properties of starch [9]. In fact, rice apparent amylose content (AC), alkali spreading value (ASV) and gel consistency (GC) are the parameters that reflect the starch properties [10]. Moreover, pasting viscosity characteristics and amylopectin structures could also be used as important starch property parameters [11,12].

Genotype, environment and their interaction could influence rice starch properties. It is reported that rice starch and its properties are controlled by different alleles of major genes or QTLs [13,14,15], and they also vary in different climates and locations and in different seasons and years [16,17,18]. It is well-known that rice sensory quality is controlled by many genes concentrated in the synthesis of rice starch and protein and influenced by the physicochemical properties of starch [19,20,21]. For the smell of cooked rice, 2-acetyl-1-pyrroline (2-AP) is recognized as one of the most important compounds traditionally, which is mainly regulated by *badh2* (*betaine aldehyde dehydrogenase homologue 2*) [22]. These studies suggested that the influence of genotype × environment on rice physicochemical properties and the influence of genotype on rice sensory and aroma are significant. However, the influence of environment on rice sensory quality and differences of VOCs of the same rice genotype grown in different places have not been studied.

In China, *indica* and *japonica* rice genotypes are usually grown in the south and north, respectively. Due to overcoming the subspecies incompatibility between *indica* and *japonica*, a new *indica*–*japonica* hybrid rice with remarkable heterosis in yield was developed [23]. It has been widely promoted in Zhejiang Province, China. Zhejiang Province is located on the southeast coast of China, belonging to the subtropical monsoon climate. It is very suitable for planting *indica*, *japonica* and *indica–japonica* hybrid rice and the planting area accounts for about 25%, 35% and 40%, respectively. Although rice production remained stable at a high level, the differences in eating quality and aroma among the three rice types and how the planting environments affect them are still unclear.

In this study, in order to investigate the effects of genotype, environment and their interaction on physicochemical and sensory properties and find out the differences of VOCs among the three rice types, two *indica*, two *japonica* and two *indica–japonica* rice varieties widely planted in ten locations of Zhejiang Province were collected. The physicochemical properties, including AC, total protein content (PC), GC, ASV and RVA profiles and sensory properties, including smell, appearance, palatability, texture of cooked rice, and VOCs detected by GC-IMS, were investigated. It would help rice breeders to know how the environment affects the physicochemical and sensory properties and VOCs of the three rice types. It is also important for food enterprises to provide rice products with stable quality.

## 2. Materials and Methods

### 2.1. Rice Samples

Six rice genotypes (*Oryza sativa* L.) of three types were selected, HZY261 (Huazheyou261, *indica*), ZZY8 (Zhongzheyou8, *indica*), JFY2 (Jiafengyou2, *indica–japonica*), YY15 (Yongyou15, *indica–japonica*), JHX1 (Jiahexiang1, *japonica*) and NJ46 (Nanjing46, *japonica*). All of the genotypes were grown in 2021 at ten environments located in Zhejiang Province within the east longitude from 118.6 to 121.2° and the northern latitude from 27.8 to 30.8° (Appendix A). E1, E2, E3, E4, E5, E6, E7, E8, E9 and E10 indicates the ten planting locations in the city of Hangzhou, Ningbo, Wenzhou, Shaoxing, Huzhou, Jiaxing, Jinhua, Quzhou, Taizhou and Lishui, respectively. All rice genotypes were sown in late May, transplanted in late June, and harvested in late September or October. The weather conditions are shown in Appendix A.

All the rice samples were air-dried and stored at room temperature for 3 months and then stored at 0–4 °C. Prior to analysis, the rice grains were de-husked and milled into white rice on a rice milling machine (Yamamoto Co., Yamagata, Japan) and then passed through a 100-mesh sieve on a Cyclone sample mill (UDY Co., Fort Collins, CO, USA).

### 2.2. Chemicals

Analytical grade methanol and ethanol, sodium hydroxide, potassium hydroxide, acetic acid, potassium iodide, sulfuric acid, potassium sulfate, copper sulfate and boric acid were purchased from Sinopharm Chemical Reagent Co., Ltd. (Shanghai, China). The hydrochloric acid standard solution with a concentration of 0.1000 mol/L was purchased from Bolinda Co., Ltd. (Shenzhen, China). The certified rice reference materials from China National Rice Research Institute (GSB 11-3875-2021, Hangzhou, China) were used as standards to test rice apparent amylose content.

### 2.3. Apparent Amylose Content

Apparent amylose content (AC) was tested by the colorimetric method according to the ISO method [24]. Each measurement was conducted in triplicate.

### 2.4. Protein Content

Protein content (PC) was measured by the Kjeldahl method by FOSS SCINO (Kjeltec 8400, Hillerød, Denmark) with a conversion coefficient of 5.95. Each measurement was conducted in triplicate.

### 2.5. Gel Consistency

Gel consistency (GC) was assayed according to Cagampang et al. [25] with minor modifications. Briefly, approximately 100.0 ± 0.2 mg rice flour (at a moisture content of 12%) was placed into 11 × 100 mm (inner diameter × length) culture tubes and wetted with 0.20 mL 95% ethanol containing 0.05% thymol blue. After shaking to suspend the starch, 2.0 mL of 0.200 mol/L, KOH was added immediately and mixed on a vortex (XK80-A, Taizhou, China). The tube with the mixture was covered with a glass marble and placed into a vigorously boiling water bath for about 8 min. To prevent the mixture from boiling out, a hair dryer could be used to blow properly on the outside of the tube. The tube was removed from the water bath, incubated for 5 min at room temperature and then cooled in an ice-water bath for 15 min. Before detecting the gel length from the bottom of the tube to the gel, the tube was laid horizontally over a ruled paper graduated in millimeters for 1 h. Each measurement was conducted in triplicate.

### 2.6. Alkali Spreading Value

The alkali spreading value (ASV) was detected by the method suggested by Little et al. [26] with some modifications. Shortly, six whole rice kernels were immersed in 10.0 mL of 1.7% KOH solution in a Petri dish and separated by a clean glass rod to avoid adhesion among different kernels. After covering the lid, the Petri dish was moved to a constant temperature incubator (30 ± 2 °C) smoothly and incubated for 23 h. The level of intactness of each grain was visually examined by trained inspectors. The numeric score could be classified into seven grades: “1” for not affected kernel, “2” for swollen kernel, “3” for swollen kernel with incomplete and wide collar, “4” for swollen kernel with complete and wide collar, “5” for split or segmented kernel with complete and wide collar, “6” for dispersed kernel with merging collar and “7” for completely dispersed and intermingled kernel [27]. Each measurement was conducted in triplicate.

### 2.7. Pasting Viscosity

Rice pasting properties were measured by a Rapid Visco Analyzer (RVA-TecMaster, Australia) [28]. About 3 g rice flour (at a moisture content of 14%) was mixed with 25 g distilled water in the RVA sample can. The RVA was run using Thermocline for Windows software (TCW3). An automatic heating and cooling cycle was used as follows: 0–1.0 min, 50 °C; 1.0–4.8 min, 50–95 °C; 4.8–7.3 min, 95 °C; 7.3–11.1 min, 95–50 °C; 11.1–12.5 min, 50 °C. The peak (PV), trough (TV) and final viscosity (FV) and their derivative parameters, breakdown (BD = PV − TV), setback (SB = FV − PV), consistency (CS = FV − TV), setback ratio (SBr = FV/TV), stability (Stab = TV/PV), peak time (Ptime) and pasting temperature (PT), were recorded and calculated. Each test was determined in triplicate, and the results were expressed in cP units. Each measurement was conducted in triplicate.

### 2.8. Sensory Evaluation

Sensory evaluation was studied according to the national standard method of GB/T 15682-2008 [29]. A sensory panel from the Rice Product Quality Supervision and Inspection Center, Ministry of Agriculture and Rural Affairs (Hangzhou, China) was comprised of five professionally trained assessors (two men and three women) with more than 5 years of experience in sensory assessment of cooked rice. All the rice samples were divided into three types: *indica*, *indica–japonica* and *japonica* groups. Yuzhenxiang was used as a reference sample of *indica* and *indica–japonica* rice, and Koshihikari was used as a reference sample of *japonica* rice. For each rice type, samples were labeled, randomly shuffled and renumbered. Before evaluation, raw rice needed to be cooked by a cooker. To do so, weigh about 10 g of milled rice in a pan and quickly wash twice with about 300 mL of water at room temperature. The wash time should be controlled in 3–5 min. Then, drain the rice after washing and then soak it with water at a ratio of 1:1.2 (rice: water, *w*/*w*) for 30 min at room temperature. After the water in the steamer was boiling, put the soaked rice in it, cover the lid and steam for 40 min. Turn off the power for 20 min, open the lid of the rice cooker, and the cooked rice was prepared. Each assessor received 4 pans of cooked rice with one reference sample and three testing samples for one session. The sessions were carried out two times per day at 10:00 a.m. and 15:00 p.m. All assessors had separate sensory booths where they rinsed their taste buds with water before evaluating the next samples. Five individual attributes, smell (SCR), appearance (ACR), palatability (PCR, including hardness, viscosity and elasticity), taste of cooked rice (TCR) and texture of cold cooked rice (TCCR), were rated with the total score of 20, 20, 30, 25 and 5, respectively. The sensory evaluation value was expressed as the sum of each individual attribute. The results were expressed as the mean scores of the five assessors.

### 2.9. Characteristic Volatile Organic Compounds

According to Chen et al. [30], the volatile organic components of six varieties with the lowest and highest SCR scores in the sensory evaluation were analyzed by HS-GC-IMS consisting of a 60-position headspace auto-sampler (PAL RSI, CTC Analytics AG, Zwingen, Switzerland), an Agilent 490 Micro GC (Agilent Technologies, Santa Clara, CA, USA) and a FlavourSpec^®^ advanced IMS (G.A.S. mbH, Dortmund, Germany). An FS-SE-54-CB-1 capillary column (15 m × 0.53 mm ID, CS-Chromatographie Service, Durham, Germany) was used for chromatographic separation.

Each rice sample (2.0 g) was weighed and put into a 20 mL headspace glass sampling vial and then sealed with a silicone rubber mat. The system was flushed with 99.99% pure nitrogen at the flow rate of 2 mL/min for 2 min. After the vial was incubated at 80 °C with 500 r/min for 20 min, 500 μL of headspace mixture was automatically injected into the splitless injector by a heated syringe (85 °C). A 30 min linear gradient was operated as follows: 0–2 min, 2 mL/min; 2–10 min, 2–10 mL/min; 10–20 min, 10–100 mL/min; 20–30 min, 100–150 mL/min. The analyte was separated in the column at 60 °C and then ionized in the IMS ionization chamber at 45 °C. The flow rate of the drift gas was set at 150 mL/min. The double separation obtained in the IMS drift tube and GC column was displayed in a topographic plot, showing each feature defined by retention time, drift time and intensity value.

### 2.10. Statistical Analysis

All the parameters were determined in triplicate, and the results were expressed as means ± standard deviation (SD). Data analyses were performed with SAS version 8 software (SAS Institute, Cary, NC, USA) and R (R Foundation for Statistical Computing, Vienna, Austria). Analysis of variance (ANOVA) was carried out to determine genotypic, environmental and their interaction variations among the parameters with the general linear model procedure (PROC GLM). Differences among different rice genotypes and environments were determined by Tukey multiple comparison test at *p* < 0.05. The box plot was carried out by the package of ggplot2. The Mantel test was carried out by the packages of ggplot2, vegan, ggcor, and dplyr. The principal component analysis (PCA) was conducted by the package of factoextra.

## 3. Results

### 3.1. Genotype × Environment Effects on Physicochemical and Sensory Properties

The analyses of genotypic, environmental and their interaction effects on rice physicochemical and sensory properties are shown in Table 1. It indicated that genotype affected all the physicochemical parameters (AC, PC, GC, ASV, RVA parameters) and some sensory properties (ACR, PCR, TCCR and SEV) more than the environment (*p* < 0.01). The genotypic effects of AC, ASV, FV, BD, SB, CS, SBr, Stab and PTime accounted for more than 90% of total variances, and genotypic effects of ACR, PCR, TCCR and SEV accounted for only 75, 72, 56 and 70%, respectively (*p* < 0.001). Most of the physicochemical parameters (except for GC and Stab) and SCR were significantly affected by the environment (*p* < 0.05). It suggested that GC, Stab, ACR, PCR, TCR, TCCR and SEV of different genotypes responded equally across the ten environments. Environmental factors of PC and PV accounted for 33.28 and 11.18% of the total variance, respectively, and it accounted for less than 2% for AC, SB, CS, SBr and PTime. The effects of genotype × environment for physicochemical (AC, PC, ASV, PV, TV, FV, BD, SB, CS, SBr, Stab, PTime and PT) and sensory properties (SCR, ACR, TCCR) accounted for 0–13.18% and 13.46–26.63% of the total variance, respectively. They were all affected by the interaction of genotype and environment at significant levels of *p* < 0.001.

The range of AC, PC, GC and ASV of all rice varieties across ten environments are shown in Boxplot (Figure 1). Between the two varieties of the three rice types, AC of one variety (HZY261, YY15 and JHX1) was significantly higher than that of the other (ZZY8, JFY2 and NJ46) (*p* < 0.05). AC of NJ46, YY15 and JFY2 were stable under ten environments with ranges of 8.99–9.80, 15.03–16.06 and 13.36–15.57 g/100 g, respectively. ZZY8 and JHX1 were variable under the ten environments, with AC ranging from 13.06–17.18 and 15.7–19.16 g/100 g, respectively. For PC, ZZY8 and JHX1 had higher values than the other four varieties with JFY2 (5.80–6.82 g/100 g), NJ46 (5.86–7.42 g/100 g) and YY15 (5.80–7.45 g/100 g) more stable under ten environments. GC and ASV of the two *japonica* rice (JHX1 and NJ46) were higher than those of the two *indica* (HZY261 and ZZY8) and *indica–japonica* rice (JFY2 and YY15). *Indica* rice of ZZY8 and two *japonica* rice had stable GC and ASV, respectively (*p* < 0.05).

The RVA viscosity profile of the six rice genotypes differed under ten environments, especially for ZZY8 (Appendix A). As shown in Table 2, JHX1 (2222.6 cP) had significantly lower PV than the other genotypes (2453.7–2636.3 cP), but it had higher TV and FV (1757.9 and 2737.1 cP, respectively) than ZZY8 (1508.1 and 2380.6 cP, respectively) and JFY2 (1611.0 and 2581.6 cP, respectively) (*p* < 0.05). ZZY8 (2453.7 cP) and JFY2 (2548.7 cP) had similar PV, but their TV and FV were significantly different, with JFY2 significantly higher than ZZY8. PV, TV and FV of YY15 were significantly higher than those of ZZY8. JHX1 had similar TV as NJ46, but it had a higher FV and lower PV than NJ46. For the derivative parameters among all the genotypes, JHX1 had the lowest BD but the highest SB and median level of CS (*p* < 0.05). BD of ZZY8 (945.6 cP), JFY2 (937.8 cP) and YY15 (944.7 cP) were highest and at similar levels, but SB and CS of JFY2 (32.9 and 970.7 cP, respectively) were higher than those of ZZY8 (−73.1 and 872.5 cP, respectively). For PTime and PT, JHX1 was the highest, with values of 6.35 secs and 89.60 °C, respectively. ZZY8 (5.82 secs) and JFY2 (5.83 secs) had the lowest PTime, and YY15 (85.02 °C) and NJ46 (84.86 °C) had the lowest PT. Generally, the coefficient variations of FV, BD, CS and PTime of *indica–japonica* rice (JFY2 and YY15) were relatively low, and those of FV and CS of *japonica* rice (HZY261 and ZZY8) were relatively high. Among ten environments, the average PV and TV of E9 was 2545.8 and 1718.3 cP, respectively, which were both higher than those of E4 and E5. For average FV, E6 (2607.6) and E8 (2611.3) were higher than E4 (2476.5 cP). Meanwhile, E6 also had higher SB, CS and PT than E1.

The sensory evaluations of cooked rice of six rice varieties under ten environments are shown in Appendix A. SEV of all the genotypes ranged from 79.8 to 85.1 points, with JHX1 and others recognized as the grade of relatively good (71–80 points) and good (81–90 points), respectively. For all the sensory evaluation parameters, HZY261 was significantly higher than JHX1 (*p* < 0.05). The ACR of YY15 (*indica–japonica*) was higher than that of *japonica* rice (JHX1 and NJ46). ZZY8 had a TCR score of 21.1 points, which was significantly higher than NJ46 (20.3 points). ZZY8 and YY15 had the same values of TCCR and SEV (4.2 and 83.6 points), and both of them were higher than NJ46 (4.0 and 81.6 points). Generally, the coefficient variations of PCR of JHX1 and SEV of ZZY8, YY15 and JHX1 were relatively high. Among the ten environments, PCR, TCR and TCCR were at the same levels (*p* < 0.05), which ranged from 23.8 to 25.2, from 20.2 to 21.1 and from 4.0 to 4.2 points, respectively. SCR, ACR and SEV in E6 were higher than those in E2 (*p* < 0.05). The standard deviations of PCR in E5 (4.6) and SEV in E5 (6.2) and E9 (6.0) were relatively high.

### 3.2. Correlation Analysis

Correlation analysis among all the physicochemical and sensory properties of six rice genotypes in ten environments was carried out by the Mantel test. As shown in Figure 2, AC and PC were negatively correlated with PV and BD (*p* < 0.05) but positively correlated with PT (*p* < 0.001). Additionally, positive correlations were observed among AC, FV, SB, CS and SBr (*p* < 0.001). ASV was negatively associated with GC and BD and positively related to TV, SB, Stab and PTime (*p* < 0.01). SBr and Stab were calculated by dividing FV by TV and TV by PV, respectively. Therefore, SBr and Stab were positively correlated with FV and TV, respectively, and negatively correlated with TV and PV, respectively (*p* < 0.001). A positive correlation (*p* < 0.001) was detected between PV and BD, Stab and PTime, with the former two being negatively associated with SB, Stab, PTime and PT (*p* < 0.001), and the latter two positively correlated with TV, FV and SB (*p* < 0.01). PT refers to the critical temperature at which rice starch begins to swell irreversibly in heated water and lose its birefringence and crystallinity. It was positively associated with FV, SB, CS, SBr, Stab and PTime and negatively related to PV and BD (*p* < 0.001). The Mantel test showed that the weather in July (M7) could affect PC in rice grain, and GC, PV, FV, SB, and PT could be influenced by the weather in the middle and later filling stage (M9 and M10) (*p* < 0.05).

Pairwise correlations among all the sensory parameters (SCR, ACR, PCR, TCR, TCCR and SEV) were positive (*p* < 0.01) among all the samples. Some physicochemical indexes were related to the sensory evaluation parameters of cooked rice. PV and BD were positively related to all the sensory parameters (*p* < 0.01). In addition, SEV was positively correlated with GC and negatively correlated with PC, SB, Stab, PTime and PT (*p* < 0.05). SCR was positively associated with TV and negatively related to PT (*p* < 0.05). ACR was positively associated with GC and negatively related to Stab (*p* < 0.05). PCR and TCCR were negatively related to Stab and PTime (*p* < 0.05). TCR was negatively associated with PC, SB, Stab and PT (*p* < 0.05). The Mantel test showed that TCR and SEV could be influenced by the weather in the middle filling stage (M9) (*p* < 0.05).

### 3.3. Principal Component Analysis

The principal component analysis (PCA) was based on all of the physicochemical and sensory evaluation parameters of three rice types under ten environments (Figure 3). It indicated that the first two principal components could explain 87.52% of the total variance. The first (PC1) and the second principal component (PC2) explained 63.40 and 24.12% of the total variance, respectively. PC1 is mainly attributed to PV, BD, SB, FV and CS. PC2 represented FV, PV, CS, BD and TV. All of the samples could be divided into three groups, *indica*, *indica–japonica* and *japonica* rice types. For *indica* and *japonica* rice types, ZZY8 (*indica*, except for E6 and E7) and NJ46 (*japonica*) located in the positive PC1, and HZY261 (*indica*, except for E4 and E10) and JHX1 (*japonica*) located in the negative PC1. The four genotypes are scattered linearly in the PC2 direction. It suggested that the physicochemical and sensory properties of *indica* and *japonica* rice types were mainly controlled by genotype (PC1) and influenced by the environment (PC2). The *indica*–*japonica* rice type was relatively stable under the ten environments.

### 3.4. Volatile Organic Compounds

In order to extract the effective VOC characteristics of *indica*, *indica–japonica* and *japonica* rice type, the six rice genotypes with the lowest and highest SCR in sensory evaluations were analyzed by GC-IMS (Figure 4). A two-dimensional GC-IMS map of VOCs measured by GC-IMS is presented in Figure 4A. A total of 59 signals were detected, among which 54 signals were known VOCs screened from the database, and five were unknown (ID_1 to ID_5) (Figure 4B). Because the monomers and dimers corresponded to the same substances, all of the known VOCs could be divided into seven categories, aldehydes (12 compounds), alcohols (11 compounds), ketones (8 compounds), esters (5 compounds), alkenes (2 compounds), pyrroles (1 compound) and furans (1 compound) (Appendix A). Aldehydes, alcohols and esters were the most abundant classes of VOCs in all samples.

As shown in Figure 4B, *indica*, *japonica* and *indica–japonica* rice had 9, 6 and 19 VOCs with relatively higher content, respectively. There were five unknown VOCs with higher levels in *indica* and *indica–japonica* rice, which would be identified by GC-QTOF-MS/MS or other analytical methods. For *indica* rice, the levels of 1-octanol, 2-nonenal (E), 2-furfural, ethyl acetate and propyl hexanoate in HZY261-L were lower than those in HZY261-H, but the opposite tendencies were observed in ZZY8 with ZZY8-L being higher than ZZY8-H. A similar phenomenon also occurred in *japonica* (2-acetyyl-1-pyrroline, 2-butanone) and *indica–japonica* groups (1-pentanol, pentanal, 3-methylbutanol, methyl hexanoate). For *indica–japonica* rice, the levels of 2-heptanone, 2-octenal, n-hexanol and 2-pentylfuran in JFY2-L and YY15-L were higher than those in JFY2-H and YY15-H.

The principal components analysis of all the detectable VOCs and the top ten contributors are shown in Figure 4C. All the samples could be divided into three types (*indica*, *indica–japonica* and *japonica* rice), with the first two principal components explaining 90.5% of total variances, and the top ten contributing VOCs were ethyl acetate, butanal, 2-propanone, 2-acetyl-1-pyrroline, hexanal, pentanal, 1-pentanol, ID_5, ethyl acrylate and n-hexanol. It indicated that the characteristics of VOCs of the three rice types were different, and GC-IMS could be used as an efficient way to distinguish them. This study showed that VOCs of *indica–japonica* rice were more stable than those of *japonica* and *indica* rice. PC1 mainly represented ethyl acetate, and PC2 was mainly attributed to 2-acetyl-1-pyrroline.

## 4. Discussion

Starch, which consists of amylose and amylopectin, is the major component of rice and accounts for more than 80% of the total constituents [11]. The effect of rice starch on gelatinization and pasting properties and eating quality has been studied widely [8,12,31]. The physicochemical properties of rice starch are influenced by heredity and environment. *Wx* and *ALK* were two major genes controlling not only AC but also GC and RVA profiles [13]. It was reported that major QTL clusters closing to *Wx* and *ALK* locus could be detected in different environments, and these QTL clusters could affect eating and cooking quality [18]. In this study, it showed that AC, PC, GC, ASV and RVA profiles were affected by genotypes, which was similar to previous studies [15,17]. AC ranged from 9.0 to 19.1%, belonging to low and medium amylose, and *japonica* rice had wider genetic diversity (Figure 1). SB represents the retrogradation property of starch, and low SB means hard to retrograde. Interestingly, it showed that NJ46, YY15 and ZZY8 had negative SB (Table 2). It was thought to be mainly related to AC, and semi-waxy rice (AC ranging from 8% to 12%) had negative SB (−1031 to −326 cP) [32]. Compared to NJ46, ZZY8 and YY15 had higher AC and SB, while SB in the ten environments was at similar levels. It suggested that SB was mainly affected by genotype. In addition to the genetic effects, AC, PC, GC, ASV and pasting viscosity in different rice types were affected by the environment to different degrees (Table 2, Figure 1 and Appendix A). It might be because those different rice types were suited to different growing environments. It was reported that high seasonal temperature could affect rice quality significantly, especially for rice total starch and protein content and starch structure [16]. The weather in the middle and late filling periods of rice growth (September and October) was very important for the formation of rice quality, especially for SB and PT (*p* < 0.01) (Figure 2 and Appendix A). It was thought that the increased temperature could regulate the key regulatory factors in protein synthesis and metabolism pathway; affect the synthesis, transport, folding, and assembly process of grain protein; and lead to changes in grain storage protein [9]. All of the pasting viscosity parameters from RVA profiles were mapped at the *Wx* locus except for PV [15], and most of these parameters were associated with AC, ASV and GC (Figure 2). Therefore, the mechanism of how the environment regulated AC and pasting viscosity parameters might be the same. Besides AC, protein, lipids and post-harvest processing such as drying, storage and polishing might influence the pasting viscosity, and the mechanism of their influences on viscosity characteristics should be studied further.

SEV is a comprehensive examination of the sensory properties of cooked rice, including smell, appearance, palatability, taste and texture. SCR is the human perception of the volatiles of cooked rice, with 2-acetyl-1-pyrroline (2AP) being one of the most important compounds [33]. Although 2AP was controlled by *badh2 (betaine aldehyde dehydrogenase 2)* on chromosome 8 [22], it could be stimulated by drought and salinity conditions [34]. In this study, we found that SCR was influenced by environment and genotype × environment (Table 1). It suggested that there might be other volatile compounds associated with SCR and could be influenced by environments. Previously, it was indicated that six QTLs associated with the palatability of cooked rice were non-environment-specific [14]. In this study, it found that PCR was only affected by genotype (Table 1), which was similar to the previous study [14]. The TCCR and SEV were regulated by genotype and influenced by genotype × environment (Table 1), which was similar to the results for adhesiveness and cohesiveness [10]. It was reported that rice with high BD and low SB had a relatively soft texture and good eating quality [35]. Therefore, TCCR and SEV were positively related to BD and negatively related to SB (Figure 2). It was thought that high BD made the starch granules swell more easily, and retrogradation was mainly caused by the reordering of AM and the outermost short AP chains [36]. PCR, TCR, TCCR and SEV were related to PV, BD, Stab and PTime (*p* < 0.05) (Figure 2), which indicated that RVA profile characteristics could be used as the assistant selection parameters for rice with good eating quality [37]. Moreover, the weather in the middle and late filling periods of rice development was very important for SEV of cooked rice.

The PCA divided all the samples into three subgroups: *indica*, *indica–japonica* and *japonica* (Figure 3). The confidence circle of the *indica–japonica* group was smaller than that of the *indica* and *japonica* groups, which suggested that JFY2 and YY15 (*indica–japonica* rice) were very suitable for growing in Zhejiang Province. Therefore, the advantage of a hybrid between *indica* and *japonica* rice was not only reflected in yield [23] but also in quality and environmental adaptability.

HS-GC-IMS could be used as a useful tool to detect fungal and rice weevil infection at an early stage in rice storage [38,39]. It also could achieve quick, non-destructive and accurate analysis of fragrant rice geographical properties and flavor characteristics of rice wine during the fermentation process [30,40]. In rice, aldehydes, alcohols and esters were the most abundant classes of VOCs in all samples (Figure 4A and Appendix A), which was similar to a previous study [41]. It was thought that aldehydes and esters had a fruity flavor, while alcohol had a floral or fruity flavor [33]. The number of characteristic VOCs in *indica–japonica* rice was more than that in *indica* and *japonica* rice (Figure 4B). It suggested that the hybrid between *indica* and *japonica* rice could produce more VOCs in rice. Some aldehydes, ketones and alcohols were relatively higher in the rice with lower smell scores (Appendix A and Figure 4B). It might be because these green odor compositions could cause an unpleasant smell at high concentration levels [42] and were negatively correlated with flavor in sensory evaluations. Generally, rice smell was related to the VOCs compositions and their odor threshold and could be influenced by the planting environment [43]. There were eight categories of rice aroma consisting of green (hexanal, hexanol), minty/citrusy (octanal, nonanal), sweet/fruity/floral (2-phenylethanol), roasted/nutty (2-acetyl-1-pyrroline), sulphury/meaty (dimethyl sulphide), cooked/mushroom/musty (1-octen-3-one), fatty/metallic (2-octenal (E), 2-nonenal (E), 1-octanol) and medicine [7]. Rice VOCs were recognized as important, unique and traceable parameters, and they could be used to identify rice varieties from specific sources and assess geographical origin [6]. The PCA analysis showed that VOCs of *indica–japonica* rice were more stable than those of *japonica* and *indica* rice, with PC1 and PC2 being ethyl acetate and 2-acetyl-1-pyrroline, respectively (Figure 4C). It was reported that ethyl acetate and 2-acetyl-1-pyrroline were key aroma compounds in rice wine and aromatic rice, respectively [44,45]. *Indica–japonica* rice had higher yield and better eating quality [23], and its advantages in odor needed further study.

## 5. Conclusions

Rice physicochemical parameters and the sensory evaluation value of cooked rice could be affected by genotype, environment or their interactions. It showed that AC, PC, ASV, pasting viscosity and ACR, PCR, TCCR and SEV were mainly affected by genotype, whereas SCR was mainly affected by the environment (*p* < 0.05). The G × E effect was significant for most parameters. Therefore, rice breeders should plant or select rice breeding lines with good quality in suitable places. PV, TV and FV had a coefficient of variation within 10% and could not be used as efficient parameters to select the rice genotype planting in different environments, whereas SB could be used as one of the useful parameters. Interestingly, the weather in the middle and later filling stage of rice growth greatly influenced GC, PV, FV, SB and PT of the matured rice quality, which was related to the quality of cooked rice (*p* < 0.05). A total of 59 VOCs were found in three rice types, among which 9, 6 and 19 characteristic compounds were found in *indica*, *japonica* and *indica–japonica* rice types, respectively. It indicated that ethyl acetate, 2-acetyl-1-pyrroline, butanal, 2-propanone, hexanal, pentanal, 1-pentanol, ethyl acrylate and n-hexanol compounds and one unknown compound were the main TOP10 aroma substances affected by genotype × environment. PCA of rice physicochemical and sensory evaluation parameters and VOCs of *indica*, *japonica* and *indica–japonica* rice planted in ten environments indicated that *indica–japonica* rice was more stable across the ten environments in Zhejiang Province. Therefore, in order to obtain high-quality, fragrance and good-tasting rice products stably, better rice genotypes and suitable environments should be considered. As the same rice variety harvested from different locations could have different physicochemical characteristics and tastes, the varieties with wide regional adaptability would be helpful to its further promotion and application.

## Figures and Tables

**Figure 1 foods-12-03108-f001:**
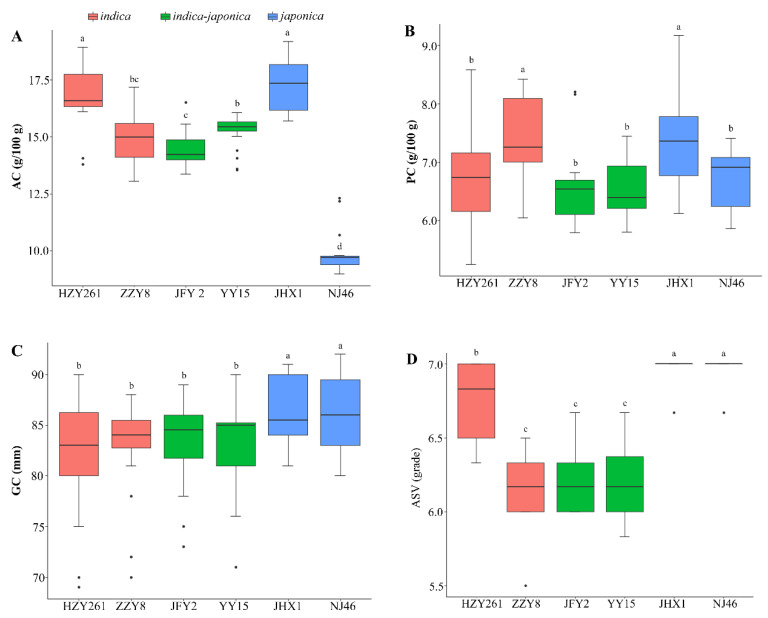
Boxplot analysis of AC, PC, GC and ASV of six rice varieties under ten environments. (**A**) Apparent amylose content (AC), (**B**) Total protein content (PC), (**C**) Gel consistency (GC), (**D**) Alkali spreading value (ASV). Boxplot analysis of AC, PC, GC and ASV of six rice varieties under ten environments. The top and bottom borders of the rectangles indicated 75th and 25th percentiles, respectively, with the median values shown in the middle of the box. Short horizontal lines above and below the rectangles meant the maximum and the minimum values, respectively, and small black dots represented outliers. Different letters above the short horizontal lines indicate significant differences (*p* < 0.05). Abbreviations are shown in Table 1.

**Figure 2 foods-12-03108-f002:**
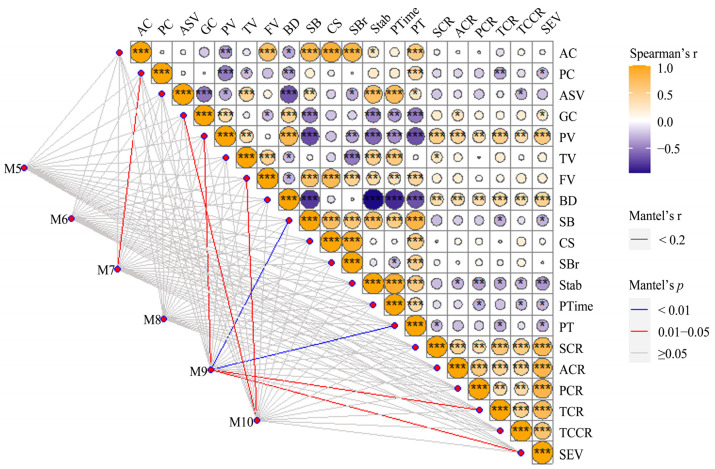
The Mantel test of rice physicochemical and sensory evaluation parameters of six rice varieties planted in ten environments. *, **, *** indicate significances at *p* = 0.05, 0.01 and 0.001 levels, respectively. M5 to M10 indicate the weather conditions in May, June, July, August, September and October, respectively. Other parameter abbreviations are shown in Table 1.

**Figure 3 foods-12-03108-f003:**
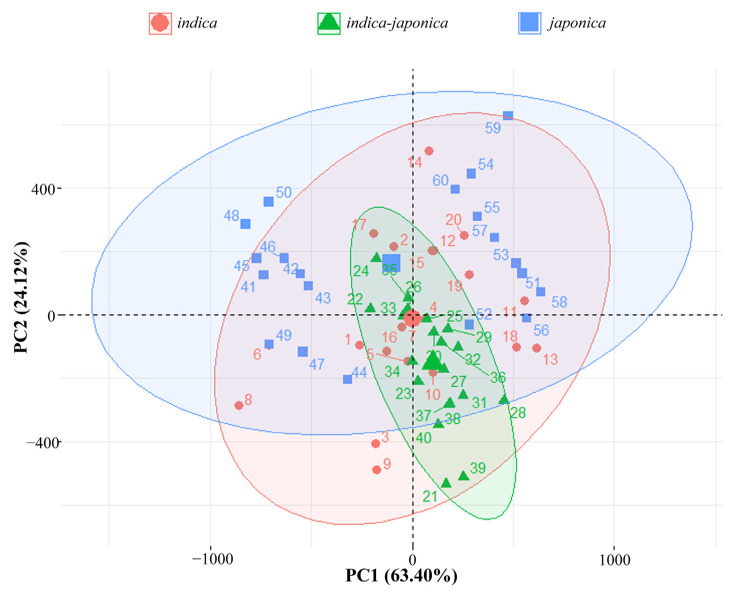
The principal component analysis of rice physicochemical and sensory evaluation parameters of six rice varieties planted in ten environments ^a^. ^a^ Sample No. 1–10: HZY261 planted in locations of E1–E10, respectively; Sample No. 11–20: ZZY8 planted in locations of E1–E10, respectively; Sample No. 21–30: JFY2 planted in locations of E1–E10, respectively; Sample No. 31–40: YY15 planted in locations of E1–E10, respectively; Sample No. 41–50: JHX1 planted in locations of E1–E10, respectively; Sample No. 51–60: NJ46 planted in locations of E1–E10, respectively.

**Figure 4 foods-12-03108-f004:**
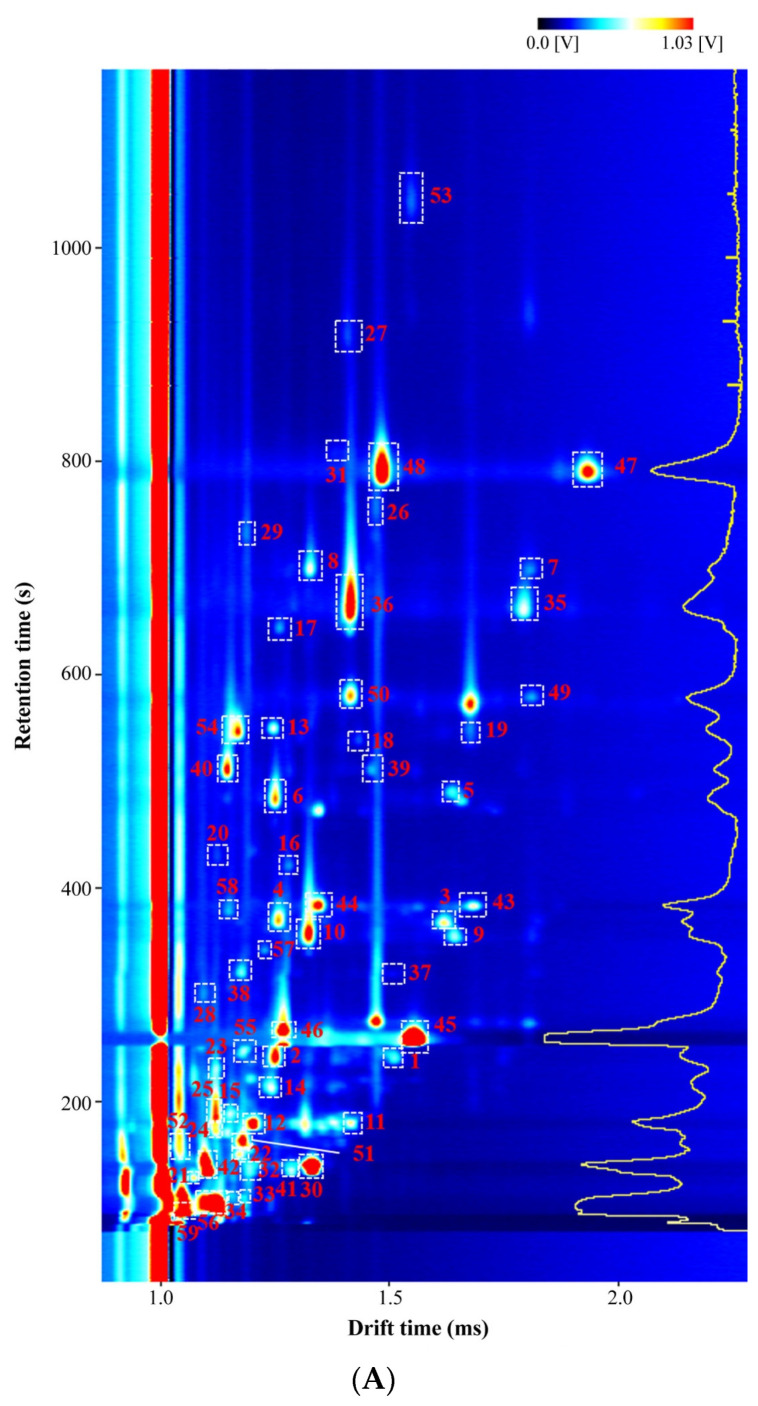
Flavor fingerprints of three rice types. (**A**) Characteristic peak selection locations from the GC-IMS topographic plots. (**B**) The selected signals by principal component analysis for three rice types. (**C**) The principal component analysis of detected signals with the top ten contributing variables. The information of signals from 1 to 59 in (**A**,**B**) are shown in Appendix A.

**Table 1 foods-12-03108-t001:** The mean square values from analysis of variances for rice physicochemical and sensory evaluation parameters ^a^.

	Genotype	Environment	Genotype × Environment
df	5	9	45
AC	138.44 ***	2.00 ***	2.31 ***
PC	3.33 ***	2.07 ***	0.82 ***
GC	63.00 **	29.14	22.81
ASV	3.39 ***	0.13 ***	0.07 ***
PV	399,211.89 ***	57,304.47 ***	55,995.30 ***
TV	173,974.05 ***	17,034.43 ***	23,336.27 ***
FV	631,332.05 ***	26,787.22 ***	32,768.32 ***
BD	688,496.63 ***	14,815.58 ***	22,161.57 ***
SB	1,395,159.39 ***	24,517.35 ***	48,816.35 ***
CS	635,183.62 ***	9858.24 ***	11,318.54 ***
SBr	0.27 ***	0.01 ***	0.01 ***
Stab	0.08 ***	0.00	0.00 ***
PTime	0.77 ***	0.01 *	0.02 ***
PT	69.44 ***	7.21 ***	5.10 ***
SCR	92.92	111.76 *	98.60 ***
ACR	321.33 ***	51.33	57.94 **
PCR	351.72 ***	58.51	77.48
TCR	97.13	41.38	44.22
TCCR	207.16 ***	62.93	98.04 ***
SEV	168.82 ***	32.15	39.54 ***

^a^ Abbreviations: df, degree of freedom; ASV, alkali spreading value; GC, gel consistency; AC, amylose content; PC, total protein content; PV, peak viscosity; TV, trough viscosity; BD, breakdown; FV, final viscosity; SB, setback; SBr, setback ratio; Stab, stability; CS, consistency; PTime, peak time; PT, pasting temperature; SCR, smell of cooked rice; ACR, appearance of cooked rice; PCR: palatability of cooked rice; TCR, taste of cooked rice; TCCR, texture of cooled cooked rice; SEV, sensitive evaluation value. *, **, *** indicate significances at *p* = 0.05, 0.01 and 0.001 levels, respectively.

**Table 2 foods-12-03108-t002:** The pasting viscosity of six rice genotypes under ten environments ^1^.

	PV (cP)	TV (cP)	FV (cP)	BD (cP)	SB (cP)	CS (cP)	PTime (secs)	PT (°C)
Genotype								
HZY261	2470.5 ± 156.3 b	1641.8 ± 118.6 bc	2711.3 ± 187.3 a	828.7 ± 123.0 b	240.8 ± 224.8 b	1069.5 ± 128.2 a	5.98 ± 0.14 c	88.41 ± 1.20 b
ZZY8	2453.7 ± 190.2 b	1508.1 ± 82.4 d	2380.6 ± 117.5 c	945.6 ± 127.7 a	–73.1 ± 193.1 d	872.5 ± 98.7 c	5.82 ± 0.08 d	86.54 ± 1.85 c
JFY2	2548.7 ± 177.9 ab	1611.0 ± 102.7 c	2581.6 ± 102.2 b	937.8 ± 100.8 a	32.9 ± 133.0 c	970.7 ± 46.3 b	5.83 ± 0.05 d	86.96 ± 1.39 c
YY15	2636.3 ± 144.0 a	1691.6 ± 104.0 ab	2628.6 ± 94.7 b	944.7 ± 69.1 a	–7.8 ± 83.3 cd	937.0 ± 32.3 b	5.93 ± 0.08 c	85.02 ± 2.25 d
JHX1	2222.6 ± 154.7 c	1757.9 ± 109.1 a	2737.1 ± 93.6 a	464.7 ± 103.3 c	514.6 ± 103.3 a	979.2 ± 45.1 b	6.35 ± 0.11 a	89.60 ± 0.97 a
NJ46	2540.0 ± 160.0 ab	1744.2 ± 124.6 a	2302.6 ± 134.9 d	795.8 ± 103.6 b	–237.5 ± 103.8 e	558.4 ± 22.6 d	6.08 ± 0.08 b	84.86 ± 2.53 d
Environment								
E1	2569.8 ± 237.8 a	1692.8 ± 113.6 ab	2588.5 ± 211.3 ab	877.0 ± 233.9 a	18.8 ± 337.6 b	895.8 ± 180.7 b	5.96 ± 0.20 a	85.98 ± 2.91 bc
E2	2418.5 ± 174.2 bc	1629.5 ± 183.7 ab	2523.8 ± 118.2 ab	789.0 ± 167.7 a	105.3 ± 215.3 ab	894.3 ± 142.9 b	6.01 ± 0.24 a	87.47 ± 1.36 ab
E3	2551.1 ± 162.8 ab	1683.6 ± 74.0 ab	2582.0 ± 191.1 ab	867.5 ± 198.2 a	30.9 ± 292.2 ab	898.4 ± 201.2 b	5.96 ± 0.22 a	86.75 ± 1.92 a–c
E4	2393.4 ± 147.5 c	1603.3 ± 153.2 b	2476.5 ± 231.5 b	790.2 ± 82.5 a	83.1 ± 151.6 ab	873.3 ± 173.1 b	5.97 ± 0.18 a	87.86 ± 1.10 a
E5	2396.3 ± 149.4 c	1614.1 ± 107.2 b	2511.5 ± 157.1 ab	782.3 ± 185.9 a	115.2 ± 252.5 ab	897.4 ± 163.3 b	6.04 ± 0.20 a	87.24 ± 2.64 ab
E6	2435.3 ± 210.7 a–c	1644.4 ± 124.2 ab	2607.6 ± 138.3 a	790.9 ± 167.5 a	172.3 ± 329.9 a	963.2 ± 224.1 a	5.98 ± 0.13 a	87.71 ± 1.82 a
E7	2487.3 ± 166.9 a–c	1665.8 ± 169.0 ab	2570.8 ± 188.1 ab	821.4 ± 182.6 a	83.5 ± 225.8 ab	904.9 ± 156.3 ab	6.02 ± 0.24 a	86.31 ± 2.01 a–c
E8	2543.5 ± 287.7 ab	1692.4 ± 66.4 ab	2611.3 ± 229.0 a	851.1 ± 299.2 a	67.8 ± 459.7 ab	918.8 ± 234.2 ab	5.98 ± 0.21 a	85.59 ± 4.56 c
E9	2545.8 ± 221.4 ab	1718.3 ± 198.2 a	2587.4 ± 343.7 ab	827.5 ± 219.8 a	41.7 ± 288.1 ab	869.2 ± 174.7 b	6.01 ± 0.22 a	86.57 ± 2.46 a–c
E10	2445.3 ± 231.9 a–c	1646.8 ± 103.5 ab	2510.1 ± 176.3 ab	798.5 ± 234.9 a	64.8 ± 242.5 ab	863.3 ± 159.9 b	6.03 ± 0.22 a	87.49 ± 1.56 ab

^1^ The results are presented as mean ± standard deviation, and values in each column of genotype or environment with different letters are significantly different (*p* < 0.05). Abbreviations are shown in Table 1.

## Data Availability

The data are available from the corresponding author.

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
