# Peer review of "The Effects of Genotype × Environment on Physicochemical and Sensory Properties and Differences of Volatile Organic Compounds of Three Rice Types (Oryza sativa L.)"

_foods, 2023, doi:10.3390/foods12163108_

Round 1

Reviewer 1 Report

After reading the manuscript "Effects of Genotype × Environment on Physicochemical and Sensory Properties and Differences of Violate Organic Compounds of Three Rice types (Oryza sativa L.)", I realized that the manuscript showed in some parts the scientific rigour wanted, but in other parts I have missed it.

The authors have presented critical evaluation only in some paragraphs.

The references are not exactly current, besides  rationale and objective could be improved.

Thats why I have written some suggestions below in an attempt to improve the paper.

L.31- Do not use as keywords, terms that already appear in your title, this will not help other researchers finding your papers.

L.36- A paragraph on the worldwide importance of this food would be important 

L.37- In the Introduction I missed a more economic, social, cultural approach on the product investigated.

L. 38- It seems to me that a " respectively" fits

L.38-40- "With the develop- 38 ment of living standards, rice with good taste and smell is more favored by the majority 39 of consumers."  author/ year ? 

L.40- you address protein in this line, then change the subject and mention protein again in line 46. This thought could be improved to make the text flows better.

L.49- Check the sequence of all citations in the paper, follow Foods guidelines. "Genotype, environment and their interaction could influence rice starch properties. It is reported that rice starch and its properties are controlled by different alleles of major 50 genes or QTLs [8-12], and they also vary in different climates and locations and in different 51 seasons and years [4, 13-15]. It is well-known that rice sensory quality, controlled by many 52 genes concentrating in the process of rice starch and protein synthesis, was influence by 53 starch physicochemical properties [16-19]. 

L.68- 71-Please reorganize the paper. Some details mentioned here are materials and methods. Rationale before objectives. Make objective clearer. I suggest following your title.

L.150-  For a sensory test a lot of relevant information was not included, I suggest reading papers and improving your article. Has the project been submitted to an evaluation by a university ethics committee? Did it follow the Helsinki declaration? Please, enter the approval protocol number.

Which sensory test was performed? How many sessions were conducted ? What was the profile of the testers ? Were the analyses performed in sensory booths ? Did the assessors receive water to rinse the taste buds ?   L- 157- " cooked rice" - It needs more detais: pan ?  water ? time ? salt / condiments ?  was it offered warm ?    L.162- 164- author ? "The results could classify the rice into six grades as very 162 poor (≤ 50 points), poor (51-60 points), normal (61-70 points), relatively good (71-80 163 points), good (81-90 points) and very good (> 90 points)."

L.166- fragrance or aroma - check, please.

186- triplication or triplicate   - check, please.

L.189- Should not include that interaction effect was tested ?

L.197 -It is confusing. The sub-heading mentions only "Physicochemical properties", but there are results on sensory as well. It needs to be adjusted, the title of the table,  also presents results on the interaction. 

L 200- RVA profiles table 1 ?  I did not find it.

L.202 - "all", does not seem to me to be the best option, because later it is written "except".

L.210- You need to mention and discuss what happened  about  the environment and genotype interaction with the sensory. It is only in the table.  ACR, PCR, TCR, TCCR, SEV ?

L.261 - Table 2 has become very disconfigured, at least for me.

Suggestion:

It needs to give space between the words genotype and the numbers.The letters representing the differences among the means could be lowercase. Decrease the size of the font of the inner numbers in the table.

L.264- I found what I had been in doubt about.

That subtitle cannot be here. Table 2 has already been presented. I would insert sensory  paragraph before table 1, since table 1 has in the heading "sensory".

L.315- "principal component explained 63.40 and 24.12% " Why ? I missed this discussion. By the way, discussion close to results is much better.  In Foods you can write both together.

 L.325- I missed these numbers for figure evaluation. Could you please include them in the original paper, they seemed important to me.

L.448- what about aroma ? violate organic compounds?After improving the objective, please adjust the conclusion as well.

Minor editing of English language required

Author Response

Thank you for your valuable suggestions and comments, which have helped us greatly improve the quality of our manuscript. Those corrected issues were clarified point-by-point and explained in the responses. Please find the attachement. The changes in our manuscript were indicated in red font. 

Reviewer 2 Report

Dear Authors,

I have through your manuscript. It's a nice piece of work. Accept few typos & grammertical errors there is no other issues that hinder its acceptance. Please check carefully the typos e.g. P 1 Line 15 & 31: should be volatile instead of Violate.

Regards

Minor editing of English language required.

Author Response

Thank you very much for the summary. We are very sorry for the typos. The “violate” was corrected into “volatile” in the whole manuscript. We carefully corrected other typos and grammatical errors throughout the manuscript, and the changes were marked in red font. 

Reviewer 3 Report

The English language needs some improvements  

Author Response

Thank you for your time and comments. Those corrected issues were clarified point-by-point and explained in the responses. Please find the attachment. The changes in our manuscript were indicated in red font.

Reviewer 4 Report

The manuscript entitled "Effects of Genotype × Environment on Physicochemical and 2 Sensory Properties and Differences of Violate Organic Com- 3 pounds of Three Rice types (Oryza sativa L.)" is quite good, but requires some improvements including:

1. The urgency of this research should be added to the beginning part of the Abstract.

2. In the Introduction section, it is unclear what novelty is being offered. Therefore it is necessary to add more specific problems or the urgency of research, and several previous studies.

3. Add the references for the sub-section of 2.7. Pasting viscosity and sub-section of 2.9. Characteristic fragrance components.

4. Samples of ZZY8, YY15, and NJ46 have negative setback viscosity (SB). This rarely happens so it needs to be discussed and explained in more depth the meaning and causes.

5. Recheck references according to the journal guidelines.

 Minor editing of English language required

Author Response

Thank you for your valuable suggestions and comments, which have helped us greatly improve the quality of our manuscript. Those corrected issues were clarified point-by-point and explained in the responses. Please find teh attachment. The changes in our manuscript were indicated in red font.

Round 2

Reviewer 1 Report

 Dear authors, 

After another evaluation of the manuscript, I realized a great improvement in the quality of the paper. The authors have accepted almost all of my requests They also have improved the English, which is always useful to ask a native speaker for a final appreciation. They added more authors to better substantiate the methodology and corrected tables and graphs.   Therefore, I  recommend "Accept" the manuscript.